

# Genetic markers of inflammation may not contribute to metabolic traits in Mexican children

Neeti Vashi[1], Carolina Stryjecki[1], Jesus Peralta-Romero[2], Fernando Suarez[2], Jaime Gomez-Zamudio[2], Ana I. Burguete-Garcia[3], Miguel Cruz[2] and David Meyre[1,4]

[1] Clinical Epidemiology & Biostatistics, McMaster University, Hamilton, Canada
[2] Medical Research Unit in Biochemistry, Hospital de Especialidades, Centro Médico Nacional Siglo XXI del Instituto Mexicano del Seguro Social, Mexico City, Mexico
[3] Centro de investigación sobre enfermedades infecciosas, Instituto Nacional de Salud Pública, Cuernavaca, Mexico
[4] Department of Pathology and Molecular Medicine, McMaster University, Hamilton, Ontario, Canada

Corresponding author
David Meyre, meyred@mcmaster.ca

## ABSTRACT

**Background:** Low-grade chronic inflammation is a common feature of obesity and its cardio-metabolic complications. However, little is known about a possible causal role of inflammation in metabolic disorders. Mexico is among the countries with the highest obesity rates in the world and the admixed Mexican population is a relevant sample due to high levels of genetic diversity.

**Methods:** Here, we studied 1,462 Mexican children recruited from Mexico City. Six genetic variants in five inflammation-related genes were genotyped: rs1137101 (leptin receptor (*LEPR*)), rs7305618 (hepatocyte nuclear factor 1 alpha (*HNF1A*)), rs1800629 (tumor necrosis factor alpha (*TNFA*)), rs1800896, rs1800871 (interleukin-10 (*IL-10*)), rs1862513 (resistin (*RETN*)). Ten continuous and eight binary traits were assessed. Linear and logistic regression models were used adjusting for age, sex, and recruitment centre.

**Results:** We found that one SNP displayed a nominal evidence of association with a continuous trait: rs1800871 (*IL-10*) with LDL (beta = −0.068 ± 1.006, P = 0.01). Subsequently, we found one nominal association with a binary trait: rs7305618 (*HNF1A*) with family history of hypertension (odds-ratio = 1.389 [1.054–1.829], P = 0.02). However, no P-value passed the Bonferroni correction for multiple testing.

**Discussion:** Our data in a Mexican children population are consistent with previous reports in European adults in failing to demonstrate an association between inflammation-associated single nucleotide polymorphisms (SNPs) and metabolic traits.

## INTRODUCTION

Obesity has increased rapidly in prevalence over the last 30 years causing a growing public health burden at the worldwide level (*Ogden et al., 2010*). Obesity is no longer only a concern for high income countries, but is escalating in developing countries as well

(*Hossain, Kawar & El Nahas, 2007*). Even more concerning are the increasing rates of childhood obesity which have tripled over the last 30 years (*Ogden et al., 2010*). In 2011–2012, the age-adjusted prevalence of obesity in adults from the United States of America was 47.8, 42.5, 32.6 and 10.8% in non-Hispanic Blacks, Hispanics, non-Hispanic White Americans, and non-Hispanic Asians, respectively (*Ogden et al., 2014*). These discrepancies may be due to differences in diet, lifestyle, socioeconomic status and access to health care across ethnic groups. However, they may also reflect differences in the genetic susceptibility to obesity and metabolic disorders as evidenced by admixture studies (*Norden-Krichmar et al., 2014*). Twin studies have reported heritability estimates between 47–90% for body mass index (BMI) (*Elks et al., 2012*). Eleven monogenic genes and more than 140 polygenic loci have been identified to date, accounting for a modest fraction of the heritability of obesity (*Locke et al., 2015*; *Yazdi, Clee & Meyre, 2015*). Obesity is associated with cardio-metabolic complications (insulin resistance, type 2 diabetes, hypertension, dyslipidemia, cardiovascular disease) that cluster into the so-called metabolic syndrome (*Walker et al., 2012*). However, the relationship between obesity and associated complications is complex as obesity does not always convert into a metabolic syndrome (*Karelis, Brochu & Rabasa-Lhoret, 2004*; *Kramer, Zinman & Retnakaran, 2013*). Consistent with the phenotypic correlations seen in observational epidemiology, shared genetic contributions between the components of the metabolic syndrome suggest that shared molecular roots may be involved in the development of the metabolic syndrome (*Avery et al., 2011*; *van Vliet-Ostaptchouk et al., 2013*; *Vattikuti, Guo & Chow, 2012*).

Inflammation has recently been advocated as one of the pathophysiological mechanisms linking obesity to other metabolic complications (*Hotamisligil, 2006*). Inflammation can be defined as a protective response of an organism to infection and injury. This operates through initiating a healing process of pathogen killing and tissue repair to restore homeostasis at the infected and/or damaged sites (*Hotamisligil, 2006*). Normally, the inflammatory response to harmful stimuli is short-lived and once the damage is removed or neutralized, the inflammation is resolved through negative feedback mechanisms (*Hotamisligil, 2006*). However, inflammatory response that fails to regulate itself becomes chronic and is believed to set the stage for a broad range of diseases (*Hotamisligil, 2006*). Obesity and its cardio-metabolic complications are associated with low-grade chronic inflammation, characterized by abnormal cytokine production, activation of a network of inflammatory signal pathways, and new connective tissue formation (*Wellen & Hotamisligil, 2005*).

Genome-wide association and in a lesser extent candidate gene studies identified around fifty common genetic variants associated with serum inflammatory biomarker levels (e.g. C-reactive protein (CRP), soluble Intercellular Adhesion Molecule 1 (sICAM-1), interleukin-6 (IL-6) or soluble P-selectin) (*Raman et al., 2013*). Researchers then used these recently discovered genetic variants to determine whether this chronic inflammation is a cause of obesity and other metabolic disorders, or a consequence of it. Overall, Mendelian randomization experiments including gene variants in inflammation pathways did not evidence a causal role of inflammation in obesity or

type 2 diabetes (*Brunner et al., 2008*; *Rafiq et al., 2008*; *Welsh et al., 2010*) and conflicting results about a causal link between inflammation and cardiovascular disease have been reported (*Brunner et al., 2008*; *The Interleukin-6 Receptor Mendelian Randomisation Analysis (IL6R MR) Consortium, 2012*; *Raman et al., 2013*; *Varbo et al., 2013*). At this stage, more research is needed to understand the role of inflammation in the development of obesity and cardio-metabolic complications, particularly in non-European populations.

Metabolic syndrome is observed in childhood obesity, but can also develop in lean children, suggesting that obesity is a marker for the syndrome, not a cause (*Elliott et al., 2009*). Since obesity and its complications are associated with atherogenesis starting in childhood and early adulthood (*Tounian et al., 2001*; *Weiss, Bremer & Lustig, 2013*), a better understanding of the molecular mechanisms involved in the clustering of cardio-metabolic factors early in life may help to develop more efficient programs to prevent the development of metabolic syndrome.

The Mexican population is characterized by a high prevalence of obesity and metabolic complications. The 2012 National Health and Nutrition Survey indicates that 34.4 and 71.2% of the Mexican children and adults respectively are overweight or obese (*León-Mimila et al., 2013*; *Tounian et al., 2001*). This ranks Mexico among the countries with the highest obesity rates in the world (*Barquera et al., 2009*; *León-Mimila et al., 2013*). The prevalence of metabolic syndrome (ATP III criteria) in children and adolescents living in Mexico was estimated to be 20% in 2006 (*Barquera et al., 2009*; *Castillo et al., 2007*). Depending on the definitions used (American Heart Association/National Heart, Lung, and Blood Institute or the International Diabetes Federation), the prevalence of metabolic syndrome among Mexican adults ranges from 59.7–68.7% (*Castillo et al., 2007*). This exceptionally high burden of obesity and metabolic syndrome in the Mexican population is largely due to the rapid transition towards an 'obesogenic' environment characterized by a sedentary lifestyle, an increase in the consumption of sugar-sweetened beverages coupled with the recent proliferation of fast food restaurants (*Isordia-Salas et al., 2011*). However, the tremendous genetic variety and unique genetic architecture of the admixed Mexican population may partly account for a higher susceptibility to obesity and metabolic disturbances than in other populations (*Rivera et al., 2002*). Mexican populations consist of Native individuals as well as individuals of European or African descent (*Rivera et al., 2002*). The distributions and proportions of these three groups vary with the region studied however evidence shows very few true Natives remain as virtually all native groups show some degree of admixture, mainly with Europeans (*The SIGMA Type 2 Diabetes Consortium, 2014*). Thus, studying the Mexican population gives insight into the disease mechanisms of a variety of races due to the genetic diversity present in the population (*The SIGMA Type 2 Diabetes Consortium, 2014*; *Lisker, Ramírez & Babinsky, 1996*).

In this study, we assessed the association of 6 common genetic single nucleotide polymorphisms (SNPs) in 5 inflammation-related genes with 10 continuous and 8 binary metabolic traits in 1,462 children from the Mexican population. Our data do not favor an association between inflammatory processes and the development of metabolic complications.

## METHODS

### Study participants

A total of 1,462 unrelated children aged 6–14 having both genetic and phenotypic data available were included in this study. Children were randomly selected to participate in a cross-sectional study from four schools in Mexico City between July 2011 and July 2012. Anthropometric traits were assessed by a trained pediatrician. Blood samples were collected for biochemical measurements and DNA extraction. Information regarding family history of type 2 diabetes, obesity and hypertension was obtained via questionnaires. The study protocol was approved by the Mexican Social Security Institute National Committee and the Ethical Committee Board and all experiments were performed in accordance with relevant guidelines and regulations. Informed consent was obtained from both parents and the child.

### Genotyping

Genomic DNA was extracted from peripheral blood using the FLEX STAR Autogen platform (Holliston, MA, USA). The genotyping was performed using the TaqMan OpenArray Real-Time PCR System (Life Technologies, Carlsbad, CA, USA), following the manufacturer's instructions. We selected six SNPs in or near five genes that displayed redundant associations with inflammation-related traits in literature: rs1137101 (leptin receptor (*LEPR*)), rs7305618 (hepatocyte nuclear factor 1 alpha (*HNF1A*)), rs1800629 (tumor necrosis factor alpha (*TNFA*)), rs1800896, rs1800871 (interleukin-10 (*IL-10*)), rs1862513 (resistin (*RETN*)) (*Kilpelainen et al., 2010*; *Ortega et al., 2014*; *Raman et al., 2013*; *Wang et al., 2011*). The six SNPs showed no deviation from Hardy-Weinberg Equilibrium ($0.22 \leq P \leq 0.76$). The call rate for each of the 6 SNPs was comprised between 94.6 and 100% (Table S1). The two SNPs rs1800896 and rs1800871 in *IL-10* display modest linkage disequilibrium in the Mexican children study sample (r2 value = 0.239).

### Phenotyping

All participants were weighed using a digital scale (Seca, Hamburg, Germany). Height was measured with a portable stadiometer (Seca 225; Seca, Hamburg, Germany). Body mass index was calculated as weight (kg)/(height (m)$^2$) and classified as underweight, normal weight, overweight, obese according to the Centers for Disease Control and Prevention CDC 2000 references. Waist circumference (WC) and hip circumference (HC) were measured at the midpoint between the lowest rib and the iliac crest at the top of the iliac crest respectively, after a normal exhalation with children in the standing position. Systolic and diastolic blood pressure (SBP and DBP) were measured using a mercurial sphygmomanometer (ALPK2, Tokyo, Japan). Blood pressure readings were taken for each participant twice on the right arm in a sitting position with a 5 min rest between each measurement and the mean of the two readings was determined. Hypertension was defined as average measured blood pressure above the American Heart Association's recommendations (systolic $\geq 140$ mmHg or diastolic $\geq 90$ mmHg). Blood samples were obtained following a 12 hour fast and were analyzed for fasting glucose, total cholesterol (TC),

HDL-cholesterol (HDL), LDL-cholesterol (LDL) and triglycerides (TG) using the ILab 350 Clinical Chemistry System (Instrumentation Laboratory IL, Barcelona, Spain). Insulin (IU) was measured by chemiluminescence (IMMULITE, Siemens, USA). The 2003 ADA criteria for fasting plasma glucose (FPG) were used to classify children as normal (FPG < 5.6 mmol/L), as having impaired fasting glucose (IFG; FPG 5.6–6.9 mmol/L), or as having T2D (FPG > 7.0 mmol/L) (*Lisker, Ramírez & Babinsky, 1996*). Subjects with IFG or T2D were considered as having hyperglycemia. Dyslipidemia was defined as fasting TG ≥ 100 mg/dL (0–9 years of age) or TG ≥ 130 mg/dL (10–19 years of age) and/or HDL-C < 35 mg/dL and/or LDL-C ≥ 130 mg/dL, according to current recommendations (*Kalra et al., 2009*). Information regarding family history of type 2 diabetes, overweight/obesity, and hypertension was obtained via questionnaires.

## Statistical analyses

Statistical analyses were performed using SPSS (version 20). We assessed the power of our sample using QUANTO software version 1.2.4 (University of Southern California, Los Angeles, CA, USA). Non-biological outlier data were discarded. Due to the risk of blood hemolysis, fasting insulin values < 1 mIU/l were discarded from the study. The normal distribution of continuous variables was tested using the Kolmogorov-Smirnov test. All traits of interest deviated significantly from normality. Logarithmic transformations corrected the lack of normality for fasting insulin, improved the distribution of six traits (BMI, waist and HC, waist to hip ratio, TC, TG), despite still deviating from normality, and did not improve the distribution of fasting glucose, HDL and LDL cholesterol. Linear regression models were used to examine the association between the SNPs and metabolic traits. These tests were adjusted for sex, age and the recruitment centre. Genetic association studies were performed under an additive mode of inheritance for 5 out of 6 SNPs and the effect allele was the minor allele. Because only one AA homozygous carrier was identified for rs1800629 (*TNFA*), we used a dominant model instead. Two-sided P < 0.05 before Bonferroni correction were considered as nominally significant. After applying a Bonferroni's correction for multiple testing (18 binary/continuous traits × 6 SNPs), P-values < $4.6 \times 10^{-4}$ (0.05/108) was considered as significant.

# RESULTS

## Characteristics of the Mexican children population

The main anthropometric and biological characteristics of the 1,462 Mexican children are summarized in Table 1. Fifty-three percent of the population were males. Children exhibited an average age and BMI of 9.24 ± 2.07 years and 19.65 ± 4.20 kg/m$^2$, respectively. Using the Centers for Disease Control and Prevention 2000 references, 1.4% of the children were underweight, 49.4% were normal weight, 21.3% were overweight and 27.9% were obese. Additionally, 1.5, 3.1 and 34.9% of children displayed hypertension, hyperglycemia, and dyslipidemia, respectively. A family history of overweight/obesity, type 2 diabetes or hypertension was reported for 53.0, 12.0 and 16.3% of children, respectively (Table 1). The sample size was similar for all traits except fasting insulin (data available in 78.5% of subjects) due to the phenomenon of blood hemolysis.

**Table 1 Characteristics of the Mexican children population.**

| Trait | Mean ± standard deviation | Sample size |
|---|---|---|
| Sex (% male/female) | 53.0/47.0 | 775/687 |
| Age (years) | 9.24 ± 2.07 | 1,462 |
| BMI (kg/m$^2$) | 19.65 ± 4.20 | 1,461 |
| Waist to hip ratio | 0.85 ± 0.06 | 1,417 |
| Systolic blood pressure (mmHg) | 98.58 ± 10.88 | 1,457 |
| Diastolic blood pressure (mmHg) | 66.25 ± 8.80 | 1,458 |
| Low density lipoprotein-cholesterol (mg/dl) | 102.43 ± 26.43 | 1,462 |
| High density lipoprotein-cholesterol (mg/dl) | 50.58 ± 12.82 | 1,462 |
| Total cholesterol (mg/dl) | 157.27 ± 33.53 | 1,462 |
| Triglycerides (mg/dl) | 93.67 ± 49.69 | 1,462 |
| Fasting glucose (mmol/l) | 4.57 ± 0.53 | 1,461 |
| Fasting insulin (mIU/l) | 9.10 ± 7.05 | 1,148 |
| Underweight (%) | 1.40 | 1,462 |
| Normal weight (%) | 49.40 | 1,462 |
| Overweight (%) | 21.30 | 1,462 |
| Obese (%) | 27.90 | 1,462 |
| Hypertension (%) | 1.50 | 1,452 |
| Hyperglycemia (%) | 3.10 | 1,456 |
| Dyslipidemia (%) | 34.90 | 1,457 |
| Type 2 diabetes family history (%) | 11.98 | 1,461 |
| Hypertension family history (%) | 16.29 | 1,461 |
| Overweight/obesity family history (%) | 53.05 | 1,461 |

## Association between genetic markers of inflammation and continuous metabolic traits

The associations between the six genetic variants of inflammation and 10 continuous metabolic traits are reported in Table 2. Only one SNP displayed a nominal evidence of association: rs1800871 (*IL-10*) with LDL (beta = −0.068 ± 1.006, P = 0.010).

## Association between genetic markers of inflammation and binary metabolic traits

The associations between the six genetic markers of inflammation and eight binary metabolic traits are reported in Table 3. One nominally significant association was found: rs7305618 (*HNF1A*) with family history of hypertension (1.389 [1.054–1.829] P = 0.020). No P-value was significant after Bonferroni correction for multiple testing.

## DISCUSSION

In the present study, we assessed the association of six common genetic variants in five inflammation-related genes with 10 continuous and eight binary metabolic traits in 1,462 children from the Mexican population. We only found one nominal associations between a genetic variant and the continuous traits. Subsequently, we only found two nominal

**Table 2 Association between six genetic markers of inflammation and 10 continuous metabolic traits.**

| | BMI[a] | WHR[a] | SBP[a] | DBP[a] | LDL | HDL | TC[a] | TG[a] | FG | FI[a] |
|---|---|---|---|---|---|---|---|---|---|---|
| **rs1137101** (*LEPR*) | 0.002 ± 0.007 (0.83) | −0.003 ± 0.002 (0.19) | −0.001 ± 0.004 (0.83) | 0.003 ± 0.005 (0.50) | −0.023 ± 0.980 (0.38) | −0.009 ± 0.477 (0.73) | −0.008 ± 0.007 (0.27) | −0.007 ± 0.017 (0.66) | −0.019 ± 0.020 (0.47) | −0.008 ± 0.026 (0.78) |
| **rs7305618** (*HNF1A*) | −0.003 ± 0.010 (0.76) | −0.004 ± 0.004 (0.319) | −0.006 ± 0.006 (0.31) | −0.010 ± 0.007 (0.15) | 0.011 ± 1.478 (0.69) | 0.006 ± 0.717 (0.83) | −0.006 ± 0.011 (0.57) | −0.016 ± 0.025 (0.53) | 0.010 ± 0.029 (0.71) | 0.029 ± 0.040 (0.46) |
| **rs1800629** (*TNFA*)[b] | 0.002 ± 0.017 (0.93) | 0.008 ± 0.006 (0.75) | 0.022 ± 0.010 (0.37) | 0.005 ± 0.012 (0.85) | 0.002 ± 2.394 (0.93) | 0.037 ± 1.156 (0.14) | 0.006 ± 0.019 (0.81) | 0.021 ± 0.043 (0.42) | 0.031 ± 4.619 (0.24) | 0.000 ± 0.066 (0.99) |
| **rs1800896** (*IL-10*) | −0.001 ± 0.008 (0.94) | −0.003 ± 0.003 (0.38) | 0.005 ± 0.004 (0.28) | 0.004 ± 0.006 (0.53) | −0.006 ± 1.158 (0.84) | 0.012 ± 0.566 (0.65) | −0.001 ± 0.009 (0.92) | −0.018 ± 0.020 (0.37) | 0.034 ± 0.023 (0.21) | $3.89 \times 10^{-5}$ ± 0.031 (0.99) |
| **rs1800871** (*IL-10*) | −0.005 ± 0.007 (0.49) | −0.001 ± 0.003 (0.57) | −0.004 ± 0.004 (0.33) | −0.005 ± 0.005 (0.31) | **−0.068 ± 1.006 (0.01)** | −0.011 ± 0.489 (0.67) | −0.010 ± 0.008 (0.19) | 0.010 ± 0.017 (0.56) | −0.006 ± 0.020 (0.82) | −0.024 ± 0.027 (0.37) |
| **rs1862513** (*RETN*) | −0.020 ± 0.011 (0.08) | −0.005 ± 0.004 (0.17) | −0.005 ± 0.006 (0.39) | 0.002 ± 0.008 (0.82) | 0.022 ± 1.570 (0.41) | 0.35 ± 0.765 (0.19) | 0.014 ± 0.012 (0.24) | −0.017 ± 0.027 (0.54) | 0.010 ± 0.032 (0.71) | −0.070 ± 0.042 (0.09) |

**Notes:**

Values in bold indicate P < 0.05; data are presented as beta ± standard error (P-value).

[a] Natural logarithmic transformation applied.
[b] SNP analyzed under the dominant model.

**Table 3 Association between six genetic markers of inflammation and eight binary metabolic traits.**

| | Normal weight vs. obese | Normal weight vs. overweight and obese | Hypertension | Hyperglycemia | Dyslipidemia | Type 2 diabetes family history | Hypertension family history | Overweight/ obesity family history |
|---|---|---|---|---|---|---|---|---|
| **rs1137101** (*LEPR*) | 0.052 [0.886–1.253] (0.56) | 1.004 [0.867–1.162] (0.96) | 1.415 [0.782–2.562] (0.25) | 0.774 [0.505–1.186] (0.24) | 0.995 [0.853–1.161] (0.95) | 0.883 [0.706–1.105] (0.28) | 0.861 [0.707–1.049] (0.14) | 0.925 [0.799–1.072] (0.30) |
| **rs7305618** (*HNF1A*) | 0.849 [0.652–1.106] (0.23) | 0.879 [0.704–1.097] (0.25) | 0.732 [0.263–2.038] (0.55) | 0.938 [0.485–1.816] (0.85) | 0.871 [0.688–1.102] (0.25) | 1.347 [0.988–1.837] (0.06) | **1.389 [1.054–1.829] (0.02)** | 1.050 [0.841–1.311] (0.67) |
| **rs1800629** (*TNFA*)[a] | 1.175 [0.767–1.799] (0.46) | 1.111 [0.767–1.609] (0.58) | 1.110 [0.260–4.736] (0.89) | 0.793 [0.244–2.576] (0.70) | 1.084 [0.738–1.592] (0.68) | 0.652 [0.336–1.266] (0.21) | 0.553 [0.300–1.018] (0.06) | 1.020 [0.702–1.482] (0.92) |
| **rs1800896** (*IL-10*) | 1.011 [0.825–1.239] (0.92) | 1.050 [0.883–1.249] (0.58) | 0.827 [0.393–1.743] (0.62) | 1.348 [0.834–2.180] (0.22) | 1.041 [0.868–1.238] (0.67) | 1.060 [0.813–1.383] (0.67) | 0.991 [0.785–1.253] (0.94) | 1.104 [0.927–1.314] (0.27) |
| **rs1800871** (*IL-10*) | 0.929 [0.778–1.108] (0.41) | 0.962 [0.827–1.118] (0.61) | 0.851 [0.457–1.584] (0.61) | 1.152 [0.750–1.769] (0.52) | 0.944 [0.805–1.107] (0.48) | 1.015 [0.807–1.277] (0.90) | 0.876 [0.714–1.075] (0.20) | 0.906 [0.779–1.054] (0.20) |
| **rs1862513** (*RETN*) | 0.785 [0.585–1.052] (0.11) | 0.932 [0.736–1.180] (0.56) | 1.065 [0.416–2.726] (0.90) | 0.677 [0.310–1.477] (0.33) | 0.941 [0.733–1.209] (0.64) | 1.135 [0.801–1.609] (0.48) | 0.981 [0.713–1.350] (0.91) | 0.984 [0.777–1.246] (0.89) |

**Notes:**
Values in bold indicate P value < 0.05; data are presented as beta [confidence interval] (P-value).
[a] SNP analyzed under the dominant mode.

associations between genetic variants and continuous/binary metabolic traits. No P-value resisted to a Bonferroni correction for multiple testing ($P < 4.6 \times 10^{-4}$). The number of significant P-values obtained in this experiment at the 0.05 alpha level was less than the number of associations expected by chance (~5). Overall, our negative results do not suggest an association between inflammation-associated SNPs and metabolic traits in Mexican children. This is in line with previous reports from literature, that at best suggest a possible association between inflammation and cardiovascular events (*Elliott et al., 2009*; *Fall et al., 2015*; *The Interleukin-6 Receptor Mendelian Randomisation Analysis (IL6R MR) Consortium, 2012*; *Rafiq et al., 2008*; *Raman et al., 2013*; *Varbo et al., 2013*; *Welsh et al., 2010*). Our findings are also supported by the discoveries of hypothesis-free genome-wide association studies for metabolic traits that show a limited overlap with genetic markers of inflammation to date (*Ehret et al., 2011*; *Locke et al., 2015*; *Mahajan et al., 2014*; *Raman et al., 2013*; *Willer et al., 2013*).

Power calculations on the standard trait BMI indicate that we only have a fair likelihood to identify associations at the nominal and Bonferroni corrected levels (Figs. S1 and S2). Therefore, we cannot totally exclude that the nominal associations reported here are actually true subtle positive results. For instance, we found that the rs7305618 SNP near *HNF1A* was nominally associated with a family history of hypertension. The *HNF1A* gene encodes hepatic nuclear factor 1 alpha (HNF1a), a transcription factor expressed in the liver, pancreas, gut and kidney (*Ban et al., 2002*). Mutations in the *HNF1A* gene account for approximately 70% of cases of maturity onset diabetes of the young (MODY) (*Vaxillaire et al., 1995*). *HNF1A* mutation carriers display a distinct hypertension status (*Owen et al., 2002*). HNF-1a is an essential transcription factor in nephron development and rare coding loss-of-function mutations in *HNF1A* lead to renal malformations and renal dysfunction in mice and humans (*Bingham et al., 2000*; *Malecki et al., 2005*; *Pontoglio et al., 1996*). Testing the associations of the *HNF1A* rs7305618 SNP with adult hypertension in independent studies may therefore be relevant. Similarly, the association of rs1800871 (*IL-10*) with LDL is indirectly supported by previous reports in literature. While the adenovirus-mediated gene transfer of interleukin-10 in an hyperlipidemic LDLr knock-out mouse model results in lowering of cholesterol levels and attenuation of atherogenesis, interleukin-10 deficiency in a distinct hyperlipidemic apolipoprotein E knock-out mouse model leads to an increase of LDL and atherosclerosis (*Caligiuri et al., 2003*; *Von Der Thüsen et al., 2001*). However, further studies in independent Mexican children populations are needed to confirm these nominal associations. No study in children has assessed the association of genetic markers of inflammation with metabolic traits, making any comparisons to our data difficult.

Our study has several strengths. It is the first to explore the associations of a representative list of genetic variants related to inflammation with metabolic traits in children and in the Mexican population. Additionally, we assessed diverse metabolic traits including both continuous and binary variables. Limitations of the study include an under-optimal statistical power to identify even substantial genetic effects, especially after corrections for multiple tests (Figs. S1 and S2). Additionally, the list of SNPs related to inflammation that we assessed did not include the more recent GWAS discoveries for inflammation traits (*Raman et al., 2013*). We did not assess the association of these

SNPs with intermediate inflammatory serum markers (e.g. CRP, sICAM-1, IL-6, soluble P-selectin). Finally, using ancestry informative markers to adjust for potential population stratification was not performed in this study.

In conclusion, the association study of six SNPs in five inflammation-related genes with 10 continuous and eight binary cardio-metabolic traits in 1,462 Mexican children does not suggest an association between inflammation-associated SNPs, obesity and its metabolic complications. Additional studies with larger sample sizes, a more exhaustive panel of SNPs and the availability of both inflammatory serum biomarkers and clinical traits in Mexican and other populations will provide a more definitive answer to this important research topic.

## ACKNOWLEDGEMENTS

We acknowledge all the participants of the study. We also acknowledge Hudson Reddon and Amel Lamri for their technical assistance.

### Funding

David Meyre is supported by a Tier 2 Canada Research Chair in Genetics of Obesity. Miguel Cruz was supported by the Fundación IMSS A.C. and by the National Council of Science and Technology (CONACYT-México) with the grant SALUD-2013-C01-201471 (FONSEC SSA/IMSS/ISSSTE). The funders had no role in study design, data collection and analysis, decision to publish, or preparation of the manuscript.

### Grant Disclosures

The following grant information was disclosed by the authors:
National Council of Science and Technology (CONACYT-México): SALUD-2013-C01-201471 (FONSEC SSA/IMSS/ISSSTE).

### Competing Interests

David Meyre is an Academic Editor for PeerJ.

### Author Contributions

- Neeti Vashi conceived and designed the experiments, analyzed the data, wrote the paper, prepared figures and/or tables, reviewed drafts of the paper.
- Carolina Stryjecki analyzed the data, wrote the paper.
- Jesus Peralta-Romero conceived and designed the experiments, performed the experiments, reviewed drafts of the paper.
- Fernando Suarez performed the experiments, reviewed drafts of the paper.
- Jaime Gomez-Zamudio performed the experiments, reviewed drafts of the paper.
- Ana I. Burguete-Garcia performed the experiments, reviewed drafts of the paper.
- Miguel Cruz conceived and designed the experiments, performed the experiments, contributed reagents/materials/analysis tools, reviewed drafts of the paper.
- David Meyre conceived and designed the experiments, analyzed the data, wrote the paper, prepared figures and/or tables, reviewed drafts of the paper.

## Human Ethics

The following information was supplied relating to ethical approvals (i.e., approving body and any reference numbers):

The study protocol was approved by the Mexican Social Security Institute National Committee and the Ethical Committee Board (Approval number R-2012-785-071) and all experiments were performed in accordance with relevant guidelines and regulations. Informed written consent was obtained from both parents and the child.

## Data Deposition

The raw data has been supplied as Supplemental Dataset Files.

## Supplemental Information

Supplemental information for this article can be found online at http://dx.doi.org/10.7717/peerj.2090#supplemental-information.

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
