# Peer review of "Genetic markers of inflammation may not contribute to metabolic traits in Mexican children"

_PeerJ, doi:10.7717/peerj.2090_

## Round 0.1 · original submission · Minor Revisions

The manuscript is, in general, well written. From the experimental point of view, the authors should indicate whether they have data about inflammatory serum markers. If not, they should discuss this issue and conclusions should be softened in the light of the results obtained.

Reviewer 1 ·

Basic reporting

The paper fulfills all PeerJ criteria for publication. The writing is clear, professional, unambiguous. The introduction & background provide appropriate context for the work.
The literature is well referenced & relevant. The overall structure conforms to PeerJ standards. The Figures are relevant, reasonable quality, reasonably well labelled & described. Raw data is supplied.

Experimental design

No Comments.

Validity of the findings

Per the PeerJ publishing rules, the fact that these results lack impact and novelty and are negative/inconclusive does not represent a bar to publication although they may somewhat diminish readership enthusiasm. Given the modestly suggestive associations, replication in a second cohort might have been useful although understandably this was not done. The rationale & benefit to the literature is clearly stated. The data appear to be robust, statistically sound, & controlled.

The conclusions are well stated, linked to original research question but unfortunately are not strictly limited to the supporting results. Some statements in the discussion, which are reflected in the title, are overly speculative and not supported by the results. On the other hand, limitations and defects in the study are clearly laid out in the penultimate paragraph of the discussion.

Additional comments

The meaning of these negative findings is unclear and is inconsistent with some of the sweeping generalizations in the discussion, which are mirrored in the title. The title and several statements in the discussion are not supported by the results and are somewhat misleading. To their credit, the authors clearly and unequivocally acknowledge the limitations and defects of the study in the penultimate paragraph of the discussion. However, these limitations are completely ignored in two other paragraphs which make sweeping generalizations which are unfortunately reflected in the title which is simply incorrect and should be changed to something more modest. It appears that different parts of the manuscript are not talking to each other.

It is not clear why only these 5 genes and these 6 variants were chosen. The literature provides a plethora of inflammation-related genes which are associated with metabolic disorders including obesity and type 2 diabetes, including many from Genome Wide Association Studies, that could potentially have been selected. The rationale for the selections is not clearly stated. This study provides no evidence that inflammation is either a cause or consequence of metabolic disorders. In any case causation cannot be unequivocally established using association/correlation/regression methods.

Small candidate gene association studies, such as this one, have been unproductive in the past and have largely been superseded by high throughput next generation genotyping and sequencing methods, whether targeted at individual genes of interest, or at the level of the whole genome. Many such studies have been performed with vastly greater power than is available in this study. Given the relatively small sample size, it is difficult to make broad extrapolations from the data. The study is clearly underpowered to detect meaningful associations with metabolic traits, and the authors clearly acknowledge this. Two advantages of this study, which differentiate it from most previous studies, are that it was performed in a cohort that was (i) Mexican, and (ii) children only. In principle this is a valuable cohort for future studies if they can be more expansive in terms of the genetic variants examined.

Two key examples of some problematic, overly broad statements in the manuscript, which are not supported by the evidence, include the following (and these need to be changed):
The Title: Genetic markers of inflammation do not contribute to metabolic traits in Mexican children.
Discussion:
"These data support the hypothesis that low-grade inflammation may be seen as a consequence rather than a cause in the development of diverse metabolic disorders." This is not supported by the evidence in the paper, nor by the literature cited (much of which is association based rather than mechanistic/causal).

In general a good clearly written manuscript that needs a little restraint in the conclusions drawn. This interesting cohort should serve as a springboard to more exciting studies in the future.

Reviewer 2 ·

Basic reporting

Within this study Neeti Vashi and colleagues investigated potential associations of genetic markers of Inflammation and metabolic traits in a cohort of ~1500 Mexican children. They detected nominal associations of two variants in IL-10 and HNF1A with LDL as a continuous trait and family history of hypertension as a binary trait which did not withstand adjustment for multiple testing. Their results indicate that there is no causal link between inflammatory genetic markers and metabolic traits.

The study is novel and valid in the context of childhood obesity and the paper is well and clearly written.

The introduction is rather extensive and would benefit from shortening especially in the first part (lines 49-62) which is on more general aspects of obesity.

Experimental design

The authors tested the association of genetic markers of Inflammation with metabolic traits. However, they do not provide evidence that those genetic markers are associated with inflammatory serum parameters (hsCRP, IL-6, TNFa). Is there data from children studies showing that the association between the genetic markers and low-grade inflammation is evident? Is there any inflammatory serum parameter available in the cohort? If not please extend discussion on this point and weaken your conclusion since you do not measure inflammation in those children.

Validity of the findings

"No Comments"

Additional comments

1. Abstract
The authors mention the results after adjustment for multiple testing in the abstract but not in the results part. Please include in the results.

2. Methods
The authors state that your selection of SNPs for the study was based on "convincing" associations (line 140). Please define convincing.

3. Methods
Statistical analyses in lean and obese children are usually based on BMI percentiles and z-score rather than BMI. Did the authors use percentile-based reference data? Is the data specific for children? Please give ranges for the grouping into weight Groups.

4. Methods
Lines 178-183
The authors performed tests for normal distribution before statistical analyses and log-transformed some of the parameters accordingly. However, some of the parameters (FG, HDL, LDL) still deviated from normality. Still the authors used linear regression analyses which is based on normal distribution. Why didn't they perform non-parametric test's for these parameters? Please explain.

5. Methods
Is there data on pubertal stage and was this included as a parameter for adjustment?

6. Discussion
line 217
I think it shoul read "The number of significant P-values"

7. Table 1
What does mgl/dl stand for? Does it mean mg/dl?

---

## Round 0.2 · accepted · Accept

The authors have very satisfactorily addressed all the issues raised by the reviewers.